# The Effect of Hearing Aid Use on Cognition in Older Adults: Can We Delay Decline or Even Improve Cognitive Function?

**DOI:** 10.3390/jcm9010254

**Published:** 2020-01-17

**Authors:** Julia Sarant, David Harris, Peter Busby, Paul Maruff, Adrian Schembri, Ulrike Lemke, Stefan Launer

**Affiliations:** 1Department of Audiology and Speech Pathology, The University of Melbourne, Melbourne, Victoria 3010, Australia; pabusby@unimelb.edu.au; 2Department of Economics, The University of Melbourne, Melbourne, Victoria 3010, Australia; harris.d@unimelb.edu.au; 3CogState, Melbourne, Victoria 3000, Australia; pmaruff@cogstate.com (P.M.); aschembri@cogstate.com (A.S.); 4Sonova AG, Zurich, 8712 Stäfa, Switzerland; Ulrike.Lemke@sonova.com (U.L.); Stefan.Launer@sonova.com (S.L.)

**Keywords:** hearing loss, cognitive decline, hearing aids, older adults, age, education, sex, speech perception

## Abstract

Hearing loss is a modifiable risk factor for dementia in older adults. Whether hearing aid use can delay the onset of cognitive decline is unknown. Participants in this study (aged 62–82 years) were assessed before and 18 months after hearing aid fitting on hearing, cognitive function, speech perception, quality of life, physical activity, loneliness, isolation, mood, and medical health. At baseline, multiple linear regression showed hearing loss and age predicted significantly poorer executive function performance, while tertiary education predicted significantly higher executive function and visual learning performance. At 18 months after hearing aid fitting, speech perception in quiet, self-reported listening disability and quality of life had significantly improved. Group mean scores across the cognitive test battery showed no significant decline, and executive function significantly improved. Reliable Change Index scores also showed either clinically significant improvement or stability in executive function for 97.3% of participants, and for females for working memory, visual attention and visual learning. Relative stability and clinically and statistically significant improvement in cognition were seen in this participant group after 18 months of hearing aid use, suggesting that treatment of hearing loss with hearing aids may delay cognitive decline. Given the small sample size, further follow up is required.

## 1. Introduction

By 2050, the number of people affected by dementia is predicted to double to over 131 million [1]. In addition to the development of pharmacological preventions or treatments for dementia, modifiable risk factors need to be examined and addressed. Age-related hearing loss (hearing thresholds greater than 25 dB HL; [2]) is highly prevalent in older adults from middle age, with a doubling of incidence reported with each decade [3]. The incidence of reported hearing loss ranges from 30%–60% for people aged over 65 years to 70%–90% over 85 years [4,5,6]. Hearing loss is associated with many co-morbidities, including poorer physical health, anxiety, depression, loneliness and isolation [7,8,9,10]. Despite its high prevalence and significant negative impact on quality of life and burden to society, hearing loss is undertreated. A recent report in the U.S showed only one in seven adults aged 50 and older used hearing aids, with fewer than 1 in 20 working adults aged 50–70 years doing so [11].

A number of individual studies and meta-analyses have reported an association between hearing loss, cognitive decline and dementia, with both peripheral hearing loss and central auditory dysfunction associated with accelerated risks of cognitive decline and incident dementia (e.g., [12,13,14,15]). Hearing loss has recently been recognized as a risk factor for dementia and is estimated to account for up to 9.1% of the modifiable risk for this disease [16]. While most studies to date have focused on the association between hearing loss and cognitive decline in older adults, two recent population-based cohort studies have reported that people with the highest dementia risk were adults with the onset of hearing loss in middle age. The first of these studies reported an increased rate of dementia diagnosis before age 60 with mid-life hearing loss (adjusted hazard ratio = 1.90; [17]), while the second study observed that the strongest association between hearing loss and dementia was for adults with hearing loss diagnosed between 45 and 64 years (hazard ratio = 1.40 for incident dementia; [18]). The findings of these recent studies support the position of the Lancet Commissions [16] that the increased risk of cognitive decline associated with hearing loss does not only apply to older adults (50–70 years), and that the strongest midlife risk factor for dementia is hearing loss.

Although two-thirds of dementia risk has been identified as genetic, it is estimated that over one-third of dementia cases may be preventable through lifestyle measures such as improved education, reduced smoking and the management of hearing loss, diabetes and obesity [16]. Given the reduced quality of life and other costs for affected individuals and their families, as well as the significant financial burden for society, if the onset of functional impairment could even be delayed by only a few years for some people, this would be a significant achievement. Given the association between hearing loss and increased rate of cognitive decline, there has been developing interest in whether the remediation of hearing loss could mediate the observed risk of accelerated cognitive decline for adults with hearing loss. However, the effect of hearing aid use on cognition is still unclear, as reports to date of the effects of hearing aid use have yielded mixed results. Although some cross-sectional cohort studies have reported either improved cognition from baseline or a slower rate of decline with hearing aid use [13,19,20,21,22], others have found no significant effect [20,23,24]. Further, a narrative review concluded that although hearing aid use could impact immediate cognitive function, there was no evidence long-term usage affected long-term cognition [25]. However, the first systematic review and meta-analysis of 30 studies (including 40 samples) concluded that while people with hearing loss had poorer cognition overall when compared to those with normal hearing, those who used hearing aids had better cognition than those who remained untreated [12]. It is important to note, however, that although the size of the difference in cognitive performance was less than half in treated versus untreated samples, this effect was based on data from only four studies, with evidence of publication bias. It was suggested that after adjusting for publication bias, the effect size found may in fact have been closer to half of the original effect size (0.23). Further, the conclusion that cognition was better for treated versus untreated people was based on the results of only three studies.

While the overall conclusion of the meta-analysis of research regarding the effects of hearing aid use on cognition was that treatment of hearing loss appeared to improve cognition [12], it was noted by the authors that methodological limitations in the published research to date do not allow for conclusions about whether the relationship between hearing loss and cognitive decline is causal. Limitations in the current literature include small sample size, retrospective design, and self-report of hearing loss and/or hearing aid use (often a single question), with no measurement of either hearing aid benefit or frequency of use [5,13,19]. If hearing loss was measured objectively at baseline, which was frequently not the case, further changes in hearing were not assessed over time. Other factors known to affect cognition such as education, social participation, mood, exercise, and diet were also not considered in some studies. With regard to cognition, further limitations include limited assessment of cognitive abilities, for example through the use of bedside dementia rating instruments, which are insensitive, or only one cognitive task assessed, and/or assessments administered using verbal instructions, which introduces a confounding factor for people with hearing loss, who are disadvantaged even with mild-moderate loss [5,21,26]. Finally, a further factor that has received limited attention is sex. There is currently no consensus regarding the influence of sex on the incidence of dementia, with contradictory evidence of dementia risk for females versus males [27]. Given the reported differences in the incidence of clinical sub-types of cognitive impairment, the greater effect of the apoloipoprotein APOE E4 allele on dementia risk for females, and the fact that education, an established risk factor for dementia, is often lower for females, it has been recommended that the investigation of risk factors for cognitive decline should be conducted separately for males and females [28]. In summary, for many studies, information about change in cognitive function and the rate of such change over time was not available. The same limitation applies to change in hearing loss, with a particular lack of information about benefits from hearing aids and frequency of device use. Without this information it is not possible to examine the relation between the rate of decline in hearing and rate of cognitive decline. Without information about frequency of device use and benefit, it is not possible to know whether the treatment is effective, and therefore what the effect, if any, on cognition would be. The effects of treatment of hearing loss with hearing aids have not yet been examined in a comprehensive study that objectively assesses both hearing loss and benefits of hearing aid use over time, controlling for the effects of other factors associated with cognitive decline. Although a randomized control trial is always going to be the preferred option in terms of study design, this is not feasible for a study including people with significant hearing loss; given the significant adverse effects of greater than mild hearing loss, it is not ethical to deny treatment.

The current study examined cognition over time in a prospectively recruited cohort of older adults with hearing loss in Australia who were first time users of hearing aids. Preliminary results at 18 months in a study which will follow participants for a long time interval are presented. Gold standard audiological hearing assessments were conducted prior to hearing aid fitting and at the first follow-up interval of 18 months, and cognitive function was measured using a computerized tool and visual instructions only. Hearing aid use and both objective and subjective benefits of treatment (speech perception and ease of listening) were also assessed. Data for other factors likely to influence cognition, such as physical health, social isolation, loneliness, mood, exercise, quality of life and sex were assessed so that their effects on cognition could be considered in the analysis. When ongoing data collection in a healthy aging comparison group of older Australians with typical hearing for their age yields a sample size large enough for meaningful comparison, cognitive and other outcomes will be compared between the hearing aid users and the group who are representative of Australian older adults. The present study investigated the relationship between degree of hearing loss and the extent of cognitive impairment prior to hearing aid fitting. The effect of hearing aid use over time on cognition was also examined, in addition to the effects of hearing aid use on quality of life. This longitudinal study will provide the most rigorous and expansive evidence to date of the effects of hearing aid use on cognition for older adults with hearing loss.

## 2. Experimental Section

### 2.1. Procedures

This study was carried out in accordance with the recommendations of the Australian National Health and Medical Research Council guidelines for ethical research conduct. The study protocol was approved by the University of Melbourne Behavioural and Social Sciences Human Ethics Sub-Committee (Ethics ID: 1646925). All participants gave written informed consent in accordance with the Declaration of Helsinki.

### 2.2. Participants

Ninety-nine adults aged 60–84 years, with hearing loss (mean better ear Pure Tone Average of 0.5, 1, 2 and 4 KHz of 31 dB) and no previously diagnosed or suspected cognitive impairment, participated in this study. Seventy-one percent of participants were retired, and 67% had postgraduate tertiary education. Participants had sufficient English to be able to give informed consent, comprehend test instructions and complete questionnaires. The participants were all clients of the University of Melbourne Academic Hearing Aids Clinic and had been identified as suitable to proceed with hearing aid fitting at the time of recruitment. Table 1 shows participant audiometric and demographic information.

All participants completed a pre-operative (baseline) assessment battery comprising audiometry, speech perception testing, cognitive screening and assessment, and health, quality of life, lifestyle and ease of listening questionnaires. A subset of participants assessed at the 18 month post-operative point completed an identical assessment battery and reported on their hearing aid use.

### 2.3. Audiological Assessment

Participants were assessed at baseline by an audiologist in a sound-proof booth, as part of the standard clinical pre-fitting workup. Hearing aids were chosen during a needs discussion with participants based on type and degree of hearing loss, personal aesthetic and technological preferences and communication needs. The NAL-NL2 prescription [29] was used for all hearing aid fitting unless clients preferred otherwise. All fittings were verified with real ear insertion gain measures using the Interacoustics Affinity AC440 module and adjusted to optimize individual preferences and listening comfort. Participants attended a review appointment within two to four weeks after fitting, with further review appointments made as necessary. All participants returned for a routine 12 month follow up appointment with their managing clinician. They were evaluated again 18 months after their initial hearing aid fitting by a research team audiologist. Audiometric assessment included air and bone conduction thresholds, speech discrimination assessment, and tympanometry. Speech perception ability was assessed using both word and sentence level materials. Consonant-vowel-consonant (CVC) monosyllabic words (50 word lists; scored for words and phonemes correct) were presented at 65 dB SPL in quiet in the left ear, right ear and binaurally in the unaided condition at baseline and in the best aided condition for participants who were assessed 18 months after hearing aid fitting. Speech Reception Threshold testing (SRT) was conducted using 20 Bamford-Kowal-Bench-like sentence lists in four-talker babble background noise. The test sentence was presented at 65 dB SPL, while the noise level was adaptive; the noise level was altered depending on the score achieved for each sentence. Both the target sentence and background noise were presented 1 m in front of the participant via a single speaker in the free field. The non-test ear was masked in the unilateral listening conditions using white noise set at 30 dB above the average of the 1 and 2 kHz thresholds. Correctly repeated target words were scored for each sentence, and the mean performance score in signal to noise ratio was used to calculate the participants’ ability to perceive speech in noise for the right ear, left ear and binaurally, with the final score reflecting the signal-to-noise ratio at which 50% of the key words were correctly identified.

### 2.4. Cognitive Assessment

Screening for dementia prior to commencing in the study was conducted using the Mini Mental State Examination (MMSE; [30]), a brief bedside dementia assessment. In accordance with the current (2011) National Institute for Health and Care Excellence (NICE) guidelines, a cut-off score of 24/30 was used to identify people with cognitive impairment. 

Cognition in this study was assessed using five subtests from the CogState Cognitive Battery [31,32,33,34]. These included assessments of psychomotor function (Detection test), attention (Identification test), working memory (One Back Test), visual learning (One Card Learning test) and executive function (Groton Maze Learning test). Cognitive assessments were administered by trained audiologists both pre- and 18 months post-hearing aid fitting. The CogState Battery is a computerized test battery developed for repeated assessment of cognitive performance. The battery is highly reliable (test-retest reliability for each measure ranges between 0.84 and 0.94), facilitates minimal practice effects [31], and is relatively quick to administer (approx. 30 min, depending on ability). The CogState Battery can detect decline in cognitive function that does not affect function in everyday life over even a 6 month period [35]. It has been used extensively in supervised settings to repeatedly evaluate cognition in clinical trials, with concussed patients, and with individuals with mild cognitive impairment and dementia. In older adults, CogState measures of information processing speed, attention and memory have been shown to be highly sensitive to the cognitive dysfunction and longitudinal cognitive decline [36,37]. The battery is visually presented and is therefore highly suitable for use with people with hearing loss. Both the speed and accuracy of responses are recorded and transformed on a centralised platform to yield normalised data distributions [31,32]. 

The Groton Maze Learning Test (GML) assesses executive function, taking 5 min to administer on average. Using a maze learning paradigm, the total number of errors made when attempting to learn the same hidden pathway across five trials presented consecutively is calculated.

The Detection Test (DET) assesses psychomotor function and takes 2 min to administer on average. The participant is asked to respond “yes” when a card in the centre of the screen turns over until either 25 correct responses are obtained, or the maximum time limit (2 min) is reached (whichever occurs first). Performance speed (milliseconds) taken to complete the test is recorded.

The Identification Test (IDN) assesses visual attention and takes approximately 3 min to administer. A playing card in the center of the screen that turns over is used to create a choice reaction paradigm in which the participant must answer “yes” or “no” to the question “Is the card red?” Performance speed (milliseconds) to complete the test is recorded.

The One Card Learning Test (OCL) assesses visual learning and takes on average 5 min to administer. A playing card in the center of the screen that turns over is used to create a pattern separation paradigm in which the participant is required to answer “yes” or “no” to the question “Have you seen this card before in this test?” The test measures performance accuracy.

The One Back Test (ONB) assesses working memory and takes 3 min to administer on average. A playing card in the center of the screen turns over to create an n-back paradigm in which the participant must answer “yes” or “no” to the question “Is the previous card the same?” Both speed and accuracy of performance on this task are measured.

### 2.5. Medical Health History

A detailed health history, including medical history, was taken, (including family history of mental and other neurological illnesses), a personal health history, including smoking, current and past alcohol use, illicit drug and medication use.

### 2.6. Anxiety and Depression

The Hospital Anxiety and Depression Scale (HADS) [38] measures levels of depressive and anxiety symptoms. This tool generates ordinal data and was designed for use with people who have physical health problems. For anxiety, specificity is 0.78, with sensitivity of 0.9, and for depression, 0.79 and 0.83 respectively.

### 2.7. Health Utilities Index-3 (HUI-3) Quality of Life

Health status and health-related quality of life (HRQL) for all participants was assessed using the Health Utilities Index-3 quality of life questionnaire (HUI-3; [39]), as one means of measuring hearing aid benefit. The HUI-3 measures vision, hearing, speech, ambulation, dexterity, emotion, cognition and pain. It has been a reliable, responsive and valid measure in a number of clinical studies. Utility scores provide an overall assessment of the HRQL of patients, with 0.00 representing the state of being dead, and 1.00 representing a perfect state of health.

### 2.8. Ease of Communication/Subjective Device Benefit

Ease of communication in everyday situations and subjective benefit from hearing aids were measured using the Abbreviated Profile of Hearing Aid Benefit (APHAB; [40]), a questionnaire designed to measure self-reported auditory disability in everyday living. The scale covers hearing speech in a variety of competing contexts and different everyday sounds across four subscales: Ease of Communication, Reverberation, Background Noise and Aversiveness. An overall Global Benefit score is also provided. In terms of defining benefit on individual subscales, a change of 22 or more points for any of the EC, RV or BN subscales, or 31 points or more on the AV subscale, is interpreted as significant. For a significant ‘overall picture’ benefit to be defined, each of EC, RV and BN (AV is excluded) must improve by at least 5 points (with a 10% chance of false positive error). A criterion of benefit of at least 10 points reduces the chance of a false positive interpretation of significant benefit to 4%.

### 2.9. Health and Lifestyle

The International Physical Activity Questionnaire was sued to assess health and lifestyle (IPAQ; long form, 31 items; [41]). The IPAQ monitors population levels of physical activity and inactivity in adults. It has four domains: (1) during transportation, (2) at work, (3) during household and gardening tasks and (4) during leisure time, including exercise and sport participation.

### 2.10. Loneliness and Social Participation

The Lubben Social Network Scale (LSNS; [42]) is a brief instrument designed to gauge social isolation in older adults by measuring perceived support received from family and friends. It typically takes 5–10 min to complete and comprises an equally weighted number of items used to measure size, closeness and frequency of a respondent’s social network.

The Loneliness Scale [43] is an 11-item scale designed to measure subjective feelings of loneliness as well as feelings of social isolation. Participants rate each item as either O (“I often feel this way”), S (“I sometimes feel this way”), R (“I rarely feel this way”), or N (“I never feel this way”). Typically, scale reliability in the 0.80 to 0.90 range is observed.

### 2.11. Device Use Compliance

In evaluating treatment effects, it is important to measure compliance with device use. In addition to using the data logging function of the hearing aids, all participants who were assessed at the 18 month post-fitting point completed a brief questionnaire about their use of their hearing aid/s, including how many hours per day they used it.

### 2.12. Statistical Analysis

Regression was used to quantify differences in hearing loss across education levels. The specification was
BPTA = β_0_ + β_1_ UGrad + β_2_ PGrad + U(1)
where BPTA is the PTA hearing loss in the better ear in decibels, UGrad = 1 if years of education was 13–15 years and 0 otherwise, PGrad = 1 if years of education was 16 years or greater and 0 otherwise, and U is the regression disturbance. This regression was estimated for all participants and for males and females separately. The intercept β_0_ is the average hearing for those with education levels of 12 years or fewer. The slope coefficient β_1_ is the difference between the averages of the hearing losses for those with 13–15 years education and those with 12 years or fewer. Similarly, the slope coefficient β_2_ is the difference between the averages of the hearing losses for those with 16 years or more of education and those with 12 years or fewer.

Relationships between cognition, hearing, age, education and sex were quantified at baseline using regression. For a cognition score Y (one of either GML, IDN, OCL, ONB or OCL), the regression has the form
Y = β_0_ + β_1_ BPTA + β_2_ Age + β_3_ Female + β_4_ UGrad + β_5_ PGrad + U(2)
where BPTA is the PTA hearing loss in the better ear measured in units of 10 dB, Age is age in decades, Female = 1 for a female participant and 0 otherwise. The interpretation of β_1_ is that an increase of 10 dB in hearing loss in the better ear corresponds to a change of β_1_ in the mean of Y, controlling for age, gender and years of education.

For individual participants, clinically important decline in cognition over the study period was identified using the Reliable Change Index (RCI) procedure [44,45], where a reduction in score for GML, DET, ONB, and IDN of more than 16, 0.06, 0.05, or 0.04 points respectively defines a clinically relevant improvement. An increase in those scores by more than those respective amounts defines a deterioration. OCL scores are defined in the opposite direction, with an increase of more than 0.09 defining an improvement. Changes within these ranges are not considered clinically important. Using this criterion, each participant was classified into improving, not changing or deteriorating on each of the five CogState subtests.

## 3. Results

### 3.1. Baseline

Baseline characteristics of the study sample are presented in Table 1. There was no significant difference in hearing loss between males and females, and degree of hearing loss was not significantly correlated with age (*r* = 0.01, *p* = 0.915). Almost two-thirds (64.6%) of participants had a cardiac condition, with no significant difference in proportions between males and females. Overall, 17.3% of participants had anxiety. Depression was present in only 4.1% of the participants. Loneliness was reported by 43.9% of participants, and 11.2% reported severe loneliness. Only 1 person reported feeling very socially isolated. Of note is that this was a very highly educated sample, with 66.3% of participants with postgraduate education overall. There were 57.1% of the participants who reported carrying out high levels of physical activity according to the IPAQ criterion. The mean HUI3 score for quality of life was 0.74 (quite high relative to the maximum score of 1), with no significant difference between males and females. There were 61.2% of participants who reported no hearing disability according to the HUI3 criterion, 7.1% reported mild hearing disability, 26.5% moderate and 5.1% severe. 

#### 3.1.1. Hearing Loss

The regression results for Equation (1) in Table 2 show that the extent of hearing loss varied with education level. The intercept implies an average better ear hearing loss of 36 dB for those with high school education only. The significant postgraduate indicator implies the average better ear hearing loss for other participants was reduced by 5.35 dB relative to those with high school education only. The male and female results reveal that this effect of higher education was present only for males.

#### 3.1.2. Cognition

Descriptive statistics for scores across the Cogstate battery are given in Table 3, with no significant differences between males and females. 

Pairwise associations of cognitive performance with age, hearing loss and education are shown in Table 4. Increased age was significantly correlated with poorer executive function (*r* = 0.25), as was greater hearing loss (*r* = 0.25). Participants with postgraduate education (i.e., greater than 15 years of education) had significantly better executive function performance than did those with less than 15 years of education. Average cognitive performance across the various medical and demographic factors measured was also calculated, with no differences of note.

Table 5 shows regression results for Equation (2) for each of the CogState battery and the MMSE. Most noteworthy are the results for executive function (GML). Average executive function was found to be significantly worse (i.e., higher mean GML score) with greater age and greater hearing loss, and to be significantly better for those with postgraduate education.

The statistical interpretations of the coefficients in Table 5, relative to the overall mean GML score of 56.9, are that:(a)An additional 10 dB of hearing loss was associated with an increase in average GML scores by 4.20 points (7.4% of the overall mean in Table 1), controlling for the other explanatory variables (notably age);(b)An additional 10 years of age was associated with an increase in average GML scores by 8.16 points (14.3% of the overall mean), holding the other explanatory variables constant;(c)Having more than 15 years of education was associated with a decrease in average GML scores by 11.26 points (19.8% of the overall mean) relative to having 12 years or fewer.

In addition, the average visual learning (OCL) score was greater by 0.08 points (8.2% of the mean) for those with more than 15 years of education relative to those with 12 years or fewer. The average MMSE scores were significantly better for those with more than 15 years of education, and also for females. 

### 3.2. At 18 Months Post-Hearing Aid Fitting

#### 3.2.1. Hearing Aid Use and Benefit

Self-reported adaptation and hearing aid usage statistics are shown in Table 6. Over one-third (35.3%) of participants reported adapting to hearing aid use within a week of fitting and using their devices for more than 90% of waking hours by 18 months post-fitting. A total of 44.1% of participants reported using their hearing aids more than 90% of the time by 18 months post-hearing aid fitting.

Table 7 compares self-reported hearing aid use with data logging information from the hearing aids, showing there was some over-reporting of hearing aid use. For example, 27.6% of participants both reported and were logged as using their hearing aids at least 90% of the time, while 31% of participants reported using their hearing aids 90% of the time while being logged as using them 60%–90% of the time.

Table 8 shows outcomes of hearing aid use in terms of objective speech perception scores at baseline and 18 months post-hearing aid fitting. Mean CVC word scores (measured in quiet listening conditions) increased significantly, from 85.46% at baseline to 93.78% at 18 months (*p* = 0.000). Although the mean bilateral SRT scores also improved after 18 months of hearing aid use, this difference was not statistically significant (*p* = 0.066).

Table 9 shows subjective self-reported listening benefit scores on the APHAB at 18 months after hearing aid fitting in terms of the percentage of participants who reported significantly lower listening disability according to the APHAB interpretation criteria. In terms of the individual subscales, over 54% of participants reported significantly lower listening disability across all listening scenarios except reverberation (noisy listening conditions). In terms of overall benefit, 56.8% of the group reported significant benefit (improvement of at least 10 points; 4% false positive chance) in the aided listening condition. When results for males and females were compared, a greater proportion of males reported significantly lower listening disability overall.

#### 3.2.2. Cognition

Table 10 gives statistics for CogState battery scores at baseline and at 18 months. In total, 37 participants had completed cognitive assessment at 18 month follow up to date, including 20 males and 17 females. The results demonstrate a significant improvement in average executive function (GML) over the 18 months of hearing aid use, with raw scores improving from 58.8 to 51.0 (*p* = 0.001; paired *t* test). Separate results for males and females suggest that changes of larger magnitude occurred for females.

Figure 1 shows boxplots of the mean executive function (GML) scores at baseline and at 18 months post-hearing aid fitting, and the pairwise differences between them. This illustrates the improvement (decrease) in these scores for many of the participants over the 18 months with their hearing aids.

Table 11 presents statistics on the percentages of participants with clinically relevant changes according to the RCI criterion. Across the full sample, 29.7% of participants showed an improvement in executive function, 67.6% remained unchanged, and 2.7% worsened. The improvement rates were similar for males and females. These are notable results since measurable improvements in executive function would not be expected in older people. Females in particular also showed a tendency for more participants to improve rather than deteriorate on working memory (ONB), visual attention (IDN) and visual learning (OCL).

Table 12 shows how the observed cognitive changes varied according to hearing aid use. Mean cognitive scores are presented separately for those whose hearing aid use was greater than 90% of the time and those whose hearing aid use was less than this. Larger average executive function gains (13.67 points on GML, 23.1% of baseline mean) were observed for participants using their hearing aid/s greater than 90% of the time. This differed significantly (*p* = 0.028) from the average improvement of 2.95 points for those using their hearing aid/s less than 90% of the time (5.0% of the baseline mean).

#### 3.2.3. Mood and Quality of Life

Table 13 shows a cross tabulation of outcomes for mood, social isolation and loneliness at baseline and at 18 months. Baseline outcomes are given in rows, and 18 month outcomes are given in columns. Given the relatively good mental health and connectedness of the participants in this study to date, the numbers of affected people are currently too small and there is too little spread in the data for formal statistical inference. Two participants were anxious at baseline but were not at 18 months, one was depressed but was not at 18 months, and four participants were severely lonely at baseline but not at 18 months. One participant was severely lonely at 18 months but was not at baseline.

Table 14 shows changes in quality of life on the HUI3 scale. Baseline outcomes are given in rows, and 18 month outcomes are given in columns. The overall quality of life HUI3 mean increased by 0.08 (*p* = 0.012 for the paired *t* test), which is also clinically significant. On the hearing disability scale, there were 19 participants who reported increased hearing disability (e.g., 14 who had no hearing disability at baseline but mild hearing disability at 18 months) and 10 who reported reduced hearing disability.

## 4. Discussion

### 4.1. Baseline

Prior to hearing aid fitting, no significant differences in either hearing loss or cognition were observed between males and females in this sample. Although a correlation between hearing loss and age would be expected to be observed in the general population [6], this was not evident in the current study due to the reduced age range and greater age relative to the general population. Although there is significant evidence in the literature of a negative association between cardiovascular disease and hearing loss [46], despite the fact that 64% of the participants in this study had symptoms of cardiovascular disease, there was no apparent effect of this on their hearing. Regression modelling (controlling for age, sex and education) showed that for males, degree of hearing loss was predicted by education, with those with postgraduate education less affected. This has been observed previously, probably due to greater occupational noise exposure for less educated males (see, for example, [47]).

#### Cognition

Executive function (GML) performance was significantly poorer for older participants in this study (a decrease in score by 14% of the overall mean for every 10 years of age), and for those with greater hearing loss (an additional 7.4% of the overall mean score with every additional 10 dB of hearing loss). The observed decline in mean executive function score with increased hearing loss at baseline cannot be attributed to age since this was controlled in the regression. Executive function was significantly better for participants with greater education (postgraduate education predicted an increased score by 19% of the overall mean. Post-graduate education also predicted improved visual learning (OCL) scores by 8% of the overall mean score, as well as significantly improved MMSE scores. As summarized in recent systematic reviews [12,14], a negative relationship between age-related hearing loss and cognitive function has now been widely reported in many studies, with significantly poorer executive function in particular associated with hearing loss in some studies [15,48,49].

Possible mechanisms underlying the cognitive decline associated with age-related peripheral hearing loss are currently a topic of investigation. A confounding factor of the association is that neuropathic changes and microvascular pathology associated with aging increase the risk of both hearing loss and cognitive decline. In fact, decline in not only hearing but all of our senses is associated with increased cognitive decline [50]. A recent review of the possible mechanisms of the relationship between hearing loss and cognition concluded that these are likely multiple, and include increased cognitive load, reduced social engagement and increased social isolation and depression as a result of decreased communicative ability, neuropathic degeneration, and sensory degradation with resulting brain atrophy [51]. Although the treatment of hearing loss with hearing aids is unlikely to alter the course of age-related neuropathic change and microvascular pathology, there is evidence that hearing aid use can significantly improve speech perception and communicative ability, thereby impacting social engagement, quality of interactions and relationships and reducing depression, anxiety and loneliness [52,53,54,55]. Given that social isolation is associated with physiological changes such as increased systolic blood pressure and glucocorticosteroid levels [56] and is as strong a risk factor for morbidity and mortality as other well-known risk factors such as smoking, obesity, and a sedentary lifestyle [57], it is also possible that decreased social isolation due to hearing aid use may improve physical health and thereby reduce the risk of cognitive decline. Although there are contradictory reports in the literature, some studies have documented slower rates of cognitive decline in users of hearing aids as compared with non-users [5,13,22].

### 4.2. At 18 Months Post-Hearing Aid Fitting

#### 4.2.1. Hearing Aid Use and Benefit

As previously discussed, a significant limitation of previous research investigating the effects of hearing aid use on cognition is that there has been no evidence of device use or benefit that could be used to examine the longitudinal relationship between the treatment of hearing loss and its effects. In the current study, despite the fact that, on average, the participants had only a mild-to-moderate hearing loss, a significant proportion of these reported becoming frequent users of their aids immediately after fitting (35.3% within a week). Although there was over-reporting regarding device usage when self-report and hearing aid data logging information were compared, data logging information showed that 59% of participants were using their devices at least 60%–90% of the time 18 months after fitting, a far higher than average reported rate of use [11]. There was a significant objective group improvement in speech perception in quiet, with mean bilateral SRT scores (in noise) also improving, although this improvement was not statistically significant. These results are in accord with the literature, where significant improvements in speech perception have been widely reported for users of hearing aids, with greater objective and subjective benefit in quiet listening conditions, and more limited benefit in noise [52,58,59,60]. Given both the signal and noise in the SRT assessment were presented from the same direction (in front of the listeners’ heads) the directional microphones in the hearing aids would have given limited benefit compared with a listening environment where the signal and noise are spatially separated. Subjective self-reported listening disability was also significantly lower in the aided vs. unaided condition across all APHAB listening subscales except reverberation, consistent with reports suggesting that hearing aids add distortion to the speech signal in noise, thereby making speech less intelligible [60]. Almost two-thirds of the group reported significantly reduced auditory/communication disability, with a greater proportion of males reporting significantly lower disability overall. Given the small numbers of each sex in the current sample, we will not attempt to interpret this difference.

#### 4.2.2. Cognition

At 18 months after hearing aid fitting, group mean scores across the CogState battery showed no significant change, although when the results were separated by sex, mean performance for males on psychomotor function (DET) did decline significantly. A significant improvement in cognitive function across the whole group was observed for executive function (GML), equivalent to 13.2% of the baseline mean score, that would not generally be expected in older adults. Examination of results by sex showed that executive function improved more for females than for males. Reliable Change Index scores also demonstrated either clinically significant improvement or stability in executive function for 97.3% of participants, with decline seen for only one male. Clinically significant improvement was also observed for females for working memory (ONB), visual attention (IDN) and visual learning (OCL).

The value of measuring device use was demonstrated in this study, with larger improvements in executive function equivalent to 23.1% of the baseline mean seen for participants who used their hearing aids greater than 90% of the time, whereas participants who used their aids less than 90% of the time improved significantly less (5% of the baseline mean). Females used their hearing aids more regularly and for longer than did males (56.3% vs. 33.3% of the time). A higher prevalence of daily and regular hearing aid use for females has been reported previously [61,62], with a steeply sloping audiogram for males, and low satisfaction cited as common reasons for non-regular hearing aid use. In addition, females have been reported to assign greater importance to being able to effectively communicate socially, experience greater anger and stress related to hearing loss, and experience greater problem awareness and less denial of hearing loss [63]. These differences in adjustment to hearing loss may provide females with greater motivation to use hearing aids. Greater hearing aid use may have contributed the significantly greater cognitive improvement seen in females across executive function (GML), working memory (ONB), visual learning (OCL) and visual attention (IDN).

The fact that the cohort in this study is highly educated has been noted previously, with significantly better baseline executive function found for participants in this study who had 15 years or more of education. Lower education has been identified as a modifiable risk factor for dementia, with higher cognitive reserve enabling more highly educated people to maintain cognitive function in the presence of brain pathology [16]. Given the high education level of two-thirds of the cohort, it would be reasonable to expect some additional cognitive reserve for this sample compared with the general population. Therefore, this study cohort cannot be claimed to be representative of the general population, but rather of a highly educated population. However, while a reduced incidence and/or slower rate of cognitive decline may be expected in such a cohort, it would not be expected that higher level cognition in older adults would significantly improve, as was observed, not only for executive function for both sexes but also for working memory, visual attention and visual learning in females. Further follow up of a larger sample in the future will allow a comparison of variation in outcomes across education levels.

As discussed previously, there are mixed reports in the literature regarding the effects of hearing aid use on cognition in older adults, with some studies reporting a positive effect on cognition [13,19,20,21,22], and others finding no significant effects [20,23,24]. Although the first systematic review and meta-analysis of research in this field concluded that people with treated hearing loss had better overall cognition than did those who remained untreated [12], this conclusion was based on limited results which were subject to publication bias. In addition, the many methodological limitations in the published research to date further limit the extent to which conclusions can be drawn about the efficacy of hearing aids as a treatment, not only for hearing loss, but in particular for cognition in older adults. The initial findings of the current study of significant improvement in cognition associated with hearing aid use in older adults add significantly to the literature, as the methodology of the current study addresses almost all of the previously discussed limitations in objectively examining hearing loss, hearing aid use and benefits of hearing aid use over time, while controlling for the effects of other known risk factors of cognitive decline.

#### 4.2.3. Quality of Life and Mood 

After 18 months of hearing aid use, self-reported overall quality of life was significantly improved. Although it has been shown that generic measures of quality of life do not demonstrate benefits from hearing aid use, the use of disease-specific instruments such as the HUI-3 have been shown to identify positive medium to large effects on quality of life for adults [64]. A difference in outcomes between sexes was again observed, with a greater proportion of females than males reporting improved quality of life after 18 months of hearing aid use. There were also reductions in hearing disability scores for 27% of participants, although a further 38% who reported no hearing disability at baseline were reporting mild hearing disability at follow up. Given the small number of participants in this initial sample, however, further follow up is required to confirm these results.

Although it is common for adults with hearing loss to experience poor mental health, loneliness and isolation [10,65,66], the participants in this study had relatively good mental health and social support at baseline, with very few anxious, depressed or lonely, and almost all of these conditions absent after 18 months of hearing aid use. A possible explanation for the lack of these conditions at baseline is the fact that the average degree of hearing loss for participants in this study was mild-to-moderate, and that therefore the impacts of hearing loss on mental health and loneliness were therefore more limited for many participants than for people with a greater degree of hearing disability. This has been observed in the three studies to date of changes in loneliness after intervention with hearing aids or cochlear implants, with cochlear implant recipients showing significant improvements in loneliness, but hearing aid users reporting no significant change in loneliness six months after fitting [67], and a dose effect observed in hearing aid users with greater degrees of hearing loss [54]. A further possible explanation for the good mental health of the study participants is that higher educational level has a cumulative protective effect throughout life against anxiety and depression [68,69,70]. The effect on depression is greater for females; thought to be due to the effects of increased work creativity and sense of control over the adult life course, given females traditionally have fewer socio-economic resources such as power, authority and earnings [71].

## 5. Conclusions

Despite the small sample size to date, both the observed relative stability and clinically and statistically significant improvement in cognition seen in this initial participant group after 18 months of hearing aid use are exciting and encouraging. Of interest also are the observed differences between males and females in hearing aid use and benefit to both quality of life and cognition. If these findings are confirmed with larger numbers in time, this could provide important information for clinical management of older adults of both sexes in the future. Further data collection over time, with increased participant numbers, the addition of genetic screening data and comparative data for a healthy aging ‘control’ group, will confirm whether the treatment of hearing loss with hearing aids can delay or even mitigate against the effects of cognitive decline. Instead of viewing the predicted significant worldwide increase in the incidence of dementia as a looming catastrophe, perhaps the ‘rising tide of dementia’ can be seen instead as ‘both a triumph of public health and an opportunity’ [72]. For the greater number of older adults who are now living for longer, the results of this study suggest that it may be possible, through early identification and appropriate management of hearing loss, to enjoy living to a greater age, free of dementia.

## Figures and Tables

**Figure 1 jcm-09-00254-f001:**
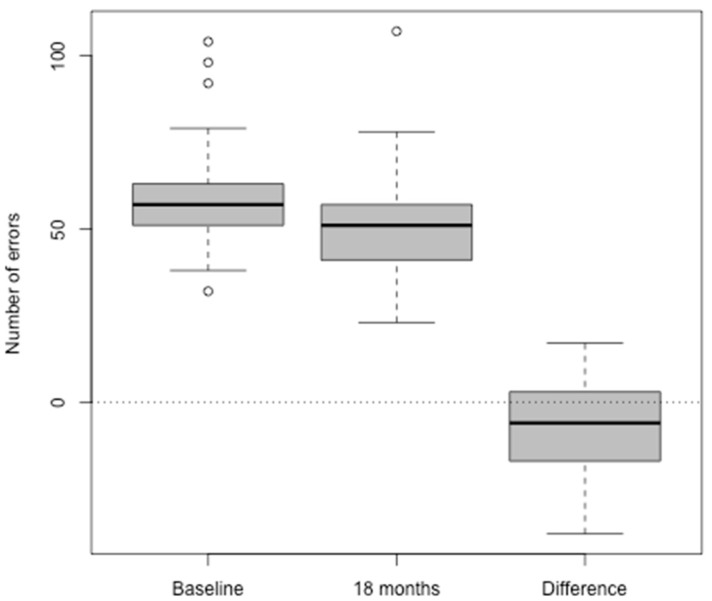
Executive function (GML) raw scores at baseline and 18 month assessments and their pairwise differences. The Y axis shows the number of errors in responses. The X axis shows assessment points and the difference in scores between the baseline and 18 month post-hearing aid fitting assessments. The boxes represent the observations between the first and third quartile. The hollow circles represent outliers. The bolded lines in the boxes represent the medians.

**Table 1 jcm-09-00254-t001:** Demographic and audiometric characteristics of participants overall and by sex at baseline (*n* = 98).

	All	Male	Female	*p*-Value *
Number	98	45	53	
Age (Mean)	72.5	73.11	71.98	
(SD)	4.86	5.01	4.73	
(Min)	62	62	62	
(Max)	83	83	82	
Better Ear PTA (Mean)	31.24	31.17	31.3	0.936
(SD)	7.9	8.51	7.42	
(Min)	15	16.25	15	
(Max)	52.5	50	52.5	
Worse Ear PTA (Mean)	38.44	40.08	37.05	0.191
(SD)	11.04	13.03	8.92	
(Min)	17.5	17.5	18.75	
(Max)	97.5	97.5	60	
Cardiac Conditions (%)	64.3	71.1	58.5	0.195
Arthritis (%)	55.1	51.1	58.5	0.47
Falls past 12 Months (%)	8.2	6.7	9.4	0.618
Diabetes (%)	7.1	6.7	7.5	0.867
High Physical Activity (%)	57.1	64.4	50.9	0.18
Retired (%)	70.4	71.1	69.8	0.89
Lonely (%)	43.9	48.9	39.6	0.363
Severely Lonely (%)	11.2	8.9	13.2	0.499
Socially Isolated (%)	1	2.2	0	0.323
Past Smoker (%)	43.9	44.4	43.4	0.918
Current Smoker (%)	3.1	2.2	3.8	0.654
HADS Depression (%)	4.1	2.2	5.7	0.38
HADS Anxiety (%)	17.3	15.6	18.9	0.668
Highest Education Level				
Up to 12 Years (%)	15.3	15.56	15.09	0.95
13–15 Years (%)	18.4	13.33	22.64	0.232
>15 Years (%)	66.3	71.11	62.26	0.358
HUI3 QoL (Mean)	0.74	0.73	0.74	0.768
HUI3 Hearing Disability (%)				
None	61.2	53.33	67.92	0.145
Mild	7.1	8.89	5.66	0.548
Moderate	26.5	31.11	22.64	0.353
Severe	5.1	6.67	3.77	0.531

*p*-Value * Significant at the 5% level; SD = standard deviation; Min = minimum value; Max = maximum value; PTA = pure tone average threshold. Physical activity was assessed using the International Physical Activity Questionnaire. Loneliness was assessed using the Loneliness Scale. Social isolation was assessed using the Lubben Social Network Scale. HADS = Hospital Anxiety and Depression Scale. HUI-3 QoL = Health Utilities Index-3 Quality of Life Measure.

**Table 2 jcm-09-00254-t002:** Variation in hearing loss with education level at baseline (*n* = 99).

	All	Male	Female
Intercept	35.75 *	37.68 *	34.06 *
	(2.00)	(3.08)	(2.64)
UGrad	−4.5	−4.97	−3.54
	(2.70)	(4.53)	(3.41)
PGrad	−5.56 *	−8.23 *	−3.15
	(2.21)	(3.40)	(2.94)

Regression coefficients with standard errors in brackets; UGrad = undergraduate education; PGrad = postgraduate education. * Significant at the 5% level.

**Table 3 jcm-09-00254-t003:** Descriptive statistics for baseline scores across the CogState battery (*n* = 98).

	GML	DET	ONB	IDN	OCL	MMSE
*n*	98	98	98	98	98	98
Mean	56.73	2.59	2.96	2.78	0.97	28.73
SD	16.9	0.09	0.08	0.06	0.12	1.43
Min	24	2.42	2.71	2.65	0.44	24
Max	119	2.8	3.17	2.95	1.23	30
Males						
*n*	45	45	45	45	45	45
Mean	53.76	2.58	2.95	2.77	0.99	28.42
SD	14.82	0.09	0.07	0.06	0.14	1.57
Min	24	2.42	2.71	2.65	0.44	24
Max	98	2.8	3.09	2.95	1.23	30
Females						
*n*	53	53	53	53	53	53
Mean	59.26	2.6	2.97	2.79	0.97	29
SD	18.24	0.08	0.09	0.06	0.1	1.24
Min	31	2.43	2.78	2.66	0.55	24
Max	119	2.78	3.17	2.93	1.13	30
Diff Means Male vs. Female						
*p*-Value	0.102	0.38	0.211	0.37	0.422	0.05

GML = executive function; DET = psychomotor function; ONB = working memory; IDN = visual attention; OCL = visual learning; MMSE = Mini Mental State Examination.

**Table 4 jcm-09-00254-t004:** Pairwise associations of cognitive performance with age, hearing loss and education at baseline.

	GML	DET	IDN	ONB	OCL	MMSE
Age Correlations						
Cor	0.25	0.2	0.04	0.11	−0.11	−0.11
*p*-Value	0.012	0.048	0.685	0.295	0.283	0.28
BPTA Correlations						
Cor	0.24	0.03	0.05	0	−0.12	−0.19
*p*-Value	0.016	0.76	0.61	0.974	0.238	0.064
Educ < 15						
N	33	33	33	33	33	33
Mean	64.15	2.6	2.78	2.95	0.94	28.18
SD	18.28	0.09	0.06	0.08	0.13	1.38
Min	33	2.47	2.66	2.84	0.44	24
Max	119	2.79	2.9	3.17	1.18	30
Educ ≥ 15						
N	65	65	65	65	65	65
Mean	52.97	2.59	2.78	2.96	0.99	29.02
SD	14.94	0.09	0.06	0.08	0.11	1.37
Min	24	2.42	2.65	2.71	0.55	24
Max	104	2.8	2.95	3.15	1.23	30
Diff Means, Educ						
*p*-Value	0.004 *	0.396	0.594	0.843	0.061	0.006 *

BPTA = Better ear pure tone average; GML = executive function; DET = psychomotor function; ONB = working memory; IDN = visual attention; OCL = visual learning; MMSE = Mini Mental State Examination; * Significant at the 5% level.

**Table 5 jcm-09-00254-t005:** Regression results at baseline for Equation (2) for each of the CogState battery subtests and the MMSE (*n* = 99).

	GML	DET	IDN	ONB	OCL	MMSE
Intercept	−9.8798	2.3531 *	2.6761 *	2.8260 *	1.0761 *	29.5289 *
	(26.1775)	(0.1449)	(0.1049)	(0.1405)	(0.1933)	(2.288)
BPTA	4.2037 *	0.0015	0.006	−0.0007	−0.0079	−0.2455
	(2.0044)	(0.0111)	(0.008)	(0.0108)	(0.0148)	(0.1752)
Age	8.1570 *	0.0343	0.0089	0.0189	−0.0182	−0.1417
	(3.2699)	(0.0181)	(0.0131)	(0.0175)	(0.0241)	(0.2858)
Educ 13–15 years	−5.6661	−0.0352	0.0125	−0.0425	0.0699	0.229
	(5.4955)	(0.0304)	(0.022)	(0.0295)	(0.0406)	(0.4803)
Educ > 15 years	11.2572 *	−0.0278	0.0178	−0.0147	0.0819 *	0.9233 *
	(4.5877)	(0.0254)	(0.0184)	(0.0246)	(0.0339)	(0.401)
Female	5.5805	0.0187	0.0125	0.0246	−0.0213	0.6324 *
	(3.1387)	(0.0174)	(0.0126)	(0.0168)	(0.0232)	(0.2743)
Rsq	0.219	0.069	0.024	0.054	0.092	0.155
*n*	99	99	99	99	99	99

BPTA = Better ear pure tone average; Coefficient estimates (standard errors in brackets) are shown for equation (2) for each subtest outcome, along with R^2^; * Significant at the 5% level.

**Table 6 jcm-09-00254-t006:** Cross tabulation of self-reported adaptation and hearing aid use (*n* = 34).

		Time to Adapt (%)			
		First Week	1 Week to 3 Months	>12 Month	>12 Months
Time worn	>90%	35.3	5.9	2.9	0.0
	60–90%	0.0	0.0	0.0	26.5
	30–60%	0.0	2.9	0.0	20.6
	<30%	0.0	0.0	0.0	5.9

**Table 7 jcm-09-00254-t007:** Hearing aid usage; cross tabulation of hearing aid data logging vs. self-report (*n* = 34).

		Reported %		
		>90%	60%–90%	30%–60%
Actual %	>90%	26.7	0	0.0
	60%–90%	30	20	0.0
	30%–60%	0	3.3	16.7
	<30%	0	0	3.3

**Table 8 jcm-09-00254-t008:** Objective speech perception scores for CVC Words and Speech Reception Thresholds (SRT) (*n* = 37).

	Mean	SD	Min	Max	Paired *t* (*p*-Value)
CVC word score	85.46	14.51	40	100	
	93.78	6.39	76	100	0.000 *
SRT score	0.23	2.1	−2.7	8.4	
	−0.39	1.06	−2.5	1.8	0.066

SRT = bilateral speech reception threshold; mean performance score in signal to noise ratio expressed in decibels and reflecting the signal-to-noise ratio at which 50% of the key words presented were correctly repeated; CVC = Consonant-vowel-consonant percent whole words correct, bilateral quiet listening condition; * Significant at the 5% level.

**Table 9 jcm-09-00254-t009:** Self-reported ease of communication (APHAB) at 18 months post-hearing aid fitting.

	Mean	SD	Min	Max	CI Lower	CI Upper	% Signif
	All	*n* = 37					
EC	13.17	9.2	1	37.5	10.11	16.24	54.1
RV	22.79	12.18	1	56	18.73	26.85	45.9
BN	24.39	15.26	1	78.7	19.3	29.48	54.1
AV	33.26	21.42	1	70.5	26.12	40.4	64.9
Global	20.11	10.64	1.6	53.8	16.56	23.66	
	Males	*n* = 20					
EC	14.32	9.7	4.7	37.5	9.79	18.86	65
RV	24.44	14.1	6.5	56	17.84	31.04	45
BN	25.43	16.9	1	78.7	17.53	33.34	60
AV	37.42	19.58	1	69.6	28.26	46.58	70
Global	21.38	11.87	7.6	53.8	15.82	26.94	
	Females	*n* = 17					
EC	11.82	8.67	1	31.2	7.36	16.27	41.2
RV	20.85	9.49	1	37.5	15.97	25.72	47.1
BN	23.16	13.49	2.8	62.3	16.23	30.1	47.1
AV	28.36	23.02	1	70.5	16.53	40.2	58.8
Global	18.61	9.12	1.6	42.3	13.92	23.3	
%	EC, RV, BN > 5	EC, RV, BN > 10					
All	75.7	56.8					
Male	80	55					
Female	70.6	58.8					

APHAB = Abbreviated Profile of Hearing Aid Benefit; EC = Ease of Communication; RV = Reverberation; BN = Background Noise; AV = Aversiveness to loud/unpleasant sounds. CI Lower and CI Upper = lower and upper bounds respectively of the 95% confidence interval for the mean.

**Table 10 jcm-09-00254-t010:** Mean difference scores at baseline and 18 months across the CogState battery for all participants and stratified by sex.

	Mean	SD	Min	Max	*p*-Value
		ALL	*n* = 37		
GML					
Baseline	58.81	15.53	32	104	
18 months	51	15.35	23	107	0.001 *
DET					
Baseline	2.58	0.08	2.44	2.75	
18 months	2.6	0.08	2.43	2.76	0.077
ONB					
Baseline	2.96	0.1	2.71	3.15	
18 months	2.94	0.08	2.81	3.18	0.205
IDN					
Baseline	2.78	0.06	2.65	2.89	
18 months	2.78	0.07	2.64	3	0.869
OCL					
Baseline	0.94	0.14	0.44	1.18	
18 months	0.96	0.11	0.6	1.1	0.262
BPTA					
Baseline	33.31	6.68	21.25	50	
18 months	35.57	7.23	23.75	55	0.001 *
		MALES	*n* = 20		
GML					
Baseline	60.3	14.02	38	98	
18 months	55.95	16.42	33	107	0.092
DET					
Baseline	2.57	0.08	2.44	2.74	
18 months	2.61	0.09	2.43	2.76	0.027 *
ONB					
Baseline	2.96	0.1	2.71	3.11	
18 months	2.93	0.08	2.81	3.13	0.117
IDN					
Baseline	2.77	0.07	2.65	2.86	
18 months	2.77	0.08	2.64	3	0.575
OCL					
Baseline	0.95	0.15	0.44	1.18	
18 months	0.96	0.1	0.71	1.1	0.643
BPTA					
Baseline	33.88	8.45	21.25	50	
18 months	36.25	8.92	23.75	55	0.003 *
		FEMALES	*n* = 17		
GML					
Baseline	57.06	17.42	32	104	
18 months	45.18	11.97	23	70	0.007 *
DET					
Baseline	2.59	0.08	2.47	2.75	
18 months	2.59	0.07	2.45	2.72	0.904
ONB					
Baseline	2.96	0.11	2.79	3.15	
18 months	2.96	0.08	2.83	3.18	0.873
IDN					
Baseline	2.8	0.05	2.7	2.89	
18 months	2.79	0.07	2.66	2.89	0.262
OCL					
Baseline	0.92	0.12	0.55	1.03	
18 months	0.95	0.11	0.6	1.1	0.229
BPTA					
Baseline	32.65	3.85	25	41.25	

BPTA = Better ear pure tone average; GML = executive function; DET = psychomotor function; ONB = working memory; IDN = visual attention; OCL = visual learning. * Significant at the 5% level.

**Table 11 jcm-09-00254-t011:** Reliable Change Index: Percentages of participants with clinically relevant changes in cognition according to the Reliable Change Index criterion.

	GML	DET	ONB	IDN	OCL
All	*n* = 37				
Improved	29.7	16.2	40.5	24.3	24.3
No Change	67.6	56.8	40.5	56.8	59.5
Improved + No Change	97.3	73	81	81.1	83.8
Declined	2.7	27	18.9	18.9	16.2
Males	*n* = 20				
Better	30	20	35	25	20
No Change	65	40	55	45	55
Better + No Change	95	60	90	70	75
Worse	5	40	10	30	25
Females	*n* = 17				
Better	29.4	11.8	47.1	23.5	29.4
No Change	70.6	76.5	23.5	70.6	64.7
Better + No Change	100	88.3	70.6	94.1	94.1
Worse	0	11.8	29.4	5.9	5.9

GML = executive function; DET = psychomotor function; ONB = working memory; IDN = visual attention; OCL = visual learning.

**Table 12 jcm-09-00254-t012:** CogState battery group mean score changes by self-reported HA usage (*n* = 37).

	>90%	≤90%	*p*
GML	−13.67	−2.95	0.028
DET	0.02	0.02	0.998
ONB	−0.03	0	0.420
IDN	0	0	0.989
OCL	0.04	0.01	0.388

GML = executive function; DET = psychomotor function; ONB = working memory; IDN = visual attention; OCL = visual learning.

**Table 13 jcm-09-00254-t013:** Cross tabulation of outcomes for mood, social isolation and loneliness at baseline and at 18 months (*n* = 33).

		18 Months	
Baseline		Not severely lonely	Severely lonely
	Not severely lonely	28	1
	Severely lonely	4	0
		Not anxious	Anxious
	Not anxious	30	0
	Anxious	2	1
		No depression	Depression
	No depression	32	0
	Depression	1	0

**Table 14 jcm-09-00254-t014:** Changes in the Health Utilities Index-3 (HUI-3) quality of life from baseline to 18 months after hearing aid fitting (*n* = 36).

			18 Months		
Baseline		None	Mild	Moderate	Severe
	None	1	14	3	0
	Mild	0	2	2	0
	Moderate	0	6	4	0
	Severe	0	2	2	0
	All	Males	Females		
Mean HUI3 Baseline	0.68	0.67	0.69		
Mean HUI3 18 months	0.76	0.74	0.78		
Paired *t* test (*p*)	* 0.012	0.125	* 0.036		

* Significant at the 5% level.

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
