# Peer review of "The Effect of Hearing Aid Use on Cognition in Older Adults: Can We Delay Decline or Even Improve Cognitive Function?"

_jcm, 2020, doi:10.3390/jcm9010254_

Round 1

Reviewer 1 Report

As the world population ages, age-related cognitive decline is becoming an increasing more prevalent and expensive burden on the health care system, economy, and individuals. The study by Sarant et al examines a simple proposition – that the use of hearing aids for older adults with hearing loss can delay cognitive decline. If true, this is a simple and cheap intervention worth considering on a large scale. The authors report that indeed their results confirm their prediction.

The data collected on each participant was impressive and thorough!

In section 2.11 it mentions a logging function of the hearing aid. how accurate, reliable, complete is that? If it provides useful data why also the need for a questionnaire? what does it add? and if the logging function is not good, how useful is it? The data from the two methods were compared in table 7 – does that mean they are redundant? compliance is a crucial element of this study.

As noted (lines 293-4; and one effect on lines 308-310 and several others on p. 9), the cohort seems to have been highly educated – can that be a confounding factor in that it may also portent less cognitive decline? That should be mentioned.

Figure 1 is missing a label on the Y-axis

Section heading 4.2 – the 4.2 should not be italics and there should not be a period after 18

Author Response

Response to Reviewer 1 Comments

Point 1: In section 2.11 it mentions a logging function of the hearing aid. how accurate, reliable, complete is that? If it provides useful data why also the need for a questionnaire? what does it add? and if the logging function is not good, how useful is it? The data from the two methods were compared in table 7 – does that mean they are redundant? compliance is a crucial element of this study.

Response 1:

The data logging function of the hearing aid provides hearing aid usage data that is objective and reliable.  However, use of a questionnaire was also included in the project methodology for two reasons:

Some hearing aids do not have a data logging function, and therefore this information is not available for some participants, who were free to choose their preferred hearing aid. Cheaper hearing aids are generally less sophisticated and may not provide a data logging function.  Given the importance of device use compliance, the only other means of obtaining this information was via a questionnaire. Use of both data logging and questionnaire responses (for those participants who were able to provide both) provides useful information about participant perceptions of their device usage, and valuable information about the reliability of this self-report information. It validated the need for objective measures of device use wherever possible, as while some participants are completely accurate in their reports, others may have a tendency to over-estimate device use.

Point 2: As noted (lines 293-4; and one effect on lines 308-310 and several others on p. 9), the cohort seems to have been highly educated – can that be a confounding factor in that it may also portent less cognitive decline? That should be mentioned.

Response 2: The cohort is highly educated compared with what would be expected for the general population.  Given the well-documented effect of education on cognition, this may indeed result in some ‘protection’, or increased cognitive reserve for these participants against cognitive decline such that lower rates of decline may be observed in this sample.  However, it would not be expected that decline could be reversed due to education coincident with the use of hearing aids.  This point has now been addressed in the discussion (lines 558-570) as follows:

“The fact that the cohort in this study is highly educated has been noted previously, with significantly better baseline executive function found for participants in this study who had 15 years or more of education.  Lower education has been identified as a modifiable risk factor for dementia, with higher cognitive reserve enabling more highly educated people to maintain cognitive function in the presence of brain pathology [16].  Given the high education level of two thirds of the cohort, it would be reasonable to expect some additional cognitive reserve for this sample compared with the general population.  Therefore this study cohort cannot be claimed to be representative of the general population, but rather of a highly educated population.  However, while a reduced incidence and/or slower rate of cognitive decline may be expected in such a cohort, it would not be expected that higher level cognition in older adults would significantly improve, as was observed, not only for executive function for both sexes but also for working memory, visual attention and visual learning in females.  Further follow up of a larger sample in the future will allow a comparison of variation in outcomes across education levels.”

Point 3: Figure 1 is missing a label on the Y-axis

 Response 3: The Y-axis has now been labelled.

Point 4: Section heading 4.2 – the 4.2 should not be italics and there should not be a period after 18

Response 4: These corrections have been made.

Reviewer 2 Report

The present manuscript describes a study on prospectively recruited 99 older adults with hearing loss regarding effects on cognition after 18 months of hearing aid (HA) usage. Both self-reported time of HA usage and logged HA usage were recorded. All subjects had no previous diagnosed or suspected cognitive impairment and were first time users of hearing aids.

Audiometric assessment included air and bone conduction and speech discrimination. Cognition was assessed by five subsets from the COGState Cognitive Battery covering psychomotor function, attention, working memory, visual learning and executive function. Further test instruments were used; testing levels of anxiety and depression, quality of life, ease of communication in everyday life, health and lifestyle, and loneliness and social participation.

After 18 months of HA usage the cognitive test battery showed no significant decline and the executive function significantly improved. Moreover there were sex differences and for females working memory, attention and visual learning were also significantly improved. It should be mentioned that speech perception in quiet, self-reported listening disability and quality of life were also significantly improved. It is concluded that the significant improvement in cognition observed suggests that treatment of hearing loss with HA may delay cognitive decline.

The perspective of an increasing population of older adults followed by an increasing frequency of dementia makes this study very interesting.  The study is well designed and carefully performed. The reliable results are extensively described in tables. The manuscript is well written with an adequate background and a good discussion.

There are however some questions and comments which mainly concern the tables which the authors should answer before a final edition is published.

- Table 2 - I am not familiar with the word “Intercept” in this context and the text to Table 2  telling  “For all participants, the intercept implies an average better ear hearing loss of 36 dB for those with high school high school education only.”  is not understandable to me. What is meant by all participants – is that all participants with high school education only?

- Table 6 - Under time to adapt (%) – what is meant by “Other”?

- Tables 13 and 14 – These tables are also difficult to read in relation to the text. How should “Baseline” be easily read in the tables?

In the last section of the Introduction the text tells that  “Preliminary results at 18 months in a study which will follow participants for a long time interval are presented”.  I imagine this study is planned to be a long-term prospective follow up of the original cohort presented here. But the results shown in this manuscript should not be defined as preliminary!  

Author Response

Response to Reviewer 2 Comments

Point 1: Table 2 - I am not familiar with the word “Intercept” in this context and the text to Table 2  telling  “For all participants, the intercept implies an average better ear hearing loss of 36 dB for those with high school high school education only.”  is not understandable to me. What is meant by all participants – is that all participants with high school education only?

Response 1:  The text has been corrected to reflect the accurate meaning of the sentence from lines 316-321 as follows:

The regression results for equation (1) in Table 2 show that the extent of hearing loss varied with education level.  The intercept implies an average better ear hearing loss of 36 dB for those with high school education only.  The significant postgraduate indicator implies the average better ear hearing loss for other participants was reduced by 5.35dB relative to those with high school education only.  The male and female results reveal that this effect of higher education was present only for males.”.

The Statistical analysis section (2.12) also now includes a representation of this regression equation as equation (1) as follows:

“Regression was used to quantify differences in hearing loss across education levels. The specification was

BPTA = β0 + β1 UGrad + β2 PGrad + U                                                                                                                                                              (1)

where BPTA is the PTA hearing loss in the better ear in decibels, UGrad = 1 if years of education was 13-15 years and 0 otherwise, PGrad = 1 if years of education was 16 years or greater and 0 otherwise, and U is the regression disturbance. This regression was estimated for all participants and for males and females separately. The intercept β0 is the average hearing for those with education levels of 12 years or fewer. The slope coefficient β1 is the difference between the averages of the hearing losses for those with 13-15 years education and those with 12 years or fewer. Similarly the slope coefficient β2 is the difference between the averages of the hearing losses for those with 16 years or more of education and those with 12 years or fewer.”

The intercept of the regression is defined in the above paragraph and is interpreted as the average hearing loss for those with high school education only. The discussion in section 3.1.1 now explicitly refers back to equation (1) and its explanation.

Table 2 has been updated to reflect these changes, as have references throughout the text to equations 1 and 2.

Point 2: Table 6 - Under time to adapt (%) – what is meant by “Other”?.

Response 2: ‘Other’ refers to greater than 12 mths taken to adapt.  This has been corrected to specify this in the table.

Point 3: Tables 13 and 14 – These tables are also difficult to read in relation to the text. How should “Baseline” be easily read in the tables?

Response 3: Tables 13 and 14 have been reformatted to make clearer the domain of the horizontal heading “Baseline”, both by vertical centering and the addition of vertical lines. The title of Table 13 now clarifies that this is a cross-tabulation table. The text preceding both tables has also been altered as follows to direct the reader as to how to read the tables:

Table 13: (lines 442-444): “Table 13 shows a cross tabulation of outcomes for mood, social isolation and loneliness at baseline and at 18 months.  Baseline outcomes are given in rows, and 18 month outcomes are given in columns.”

Table 14: (lines 453-454):

“Table 14 shows changes in quality of life on the HUI3 scale.  Baseline outcomes are given in rows, and 18 month outcomes are given in columns.”

Point 4: In the last section of the Introduction the text tells that  “Preliminary results at 18 months in a study which will follow participants for a long time interval are presented”.  I imagine this study is planned to be a long-term prospective follow up of the original cohort presented here. But the results shown in this manuscript should not be defined as preliminary!  

Response 4: The authors thank the reviewer for this complimentary comment.  However, given these results are the first report on a small sample size and for the first follow-up period of only 18 months, the authors felt it was important not to overstate the results.  If it is acceptable the reviewer, we would prefer to refer to these results as preliminary.
